# Coalescence and directed anisotropic growth of starch granule initials in subdomains of *Arabidopsis thaliana* chloroplasts

Léo Bürgy[1], Simona Eicke [1], Christophe Kopp[2], Camilla Jenny[1], Kuan Jen Lu[1], Stephane Escrig[2], Anders Meibom [2,3] & Samuel C. Zeeman[1✉]

Living cells orchestrate enzyme activities to produce myriads of biopolymers but cell-biological understanding of such processes is scarce. Starch, a plant biopolymer forming discrete, semi-crystalline granules within plastids, plays a central role in glucose storage, which is fundamental to life. Combining complementary imaging techniques and Arabidopsis genetics we reveal that, in chloroplasts, multiple starch granules initiate in stromal pockets between thylakoid membranes. These initials coalesce, then grow anisotropically to form lenticular granules. The major starch polymer, amylopectin, is synthesized at the granule surface, while the minor amylose component is deposited internally. The non-enzymatic domain of STARCH SYNTHASE 4, which controls the protein's localization, is required for anisotropic growth. These results present us with a conceptual framework for understanding the biosynthesis of this key nutrient.

[1] Institute of Molecular Plant Biology, ETH Zurich, 8092 Zurich, Switzerland. [2] Laboratory for Biological Geochemistry, Ecole Polytechnique Fédérale de Lausanne (EPFL), Lausanne, Switzerland. [3] Centre for Advanced Surface Analysis, University of Lausanne, Lausanne, Switzerland.
✉email: samuel.zeeman@biol.ethz.ch

The major source of calories for society is plant-derived carbohydrates. The most important of these carbohydrates is starch—a remarkable, insoluble biopolymer[1]. Starch forms as discrete, semi-crystalline granules within sub-cellular plastid compartments, i.e., the chloroplasts of green plants and algae and the amyloplasts of non-green heterotrophic tissues, such as seeds, roots, and tubers. The main molecular constituents of starch are the glucans amylopectin (70–90%) and amylose (10–30%). Amylopectin is a branched polymer, synthesized by a suite of interdependent enzymes, and underpins the semi-crystalline nature of starch. Amylopectin fine-structure varies between plant species as a function of the relative amounts and complex interplay between the starch biosynthesis enzymes[2]. Amylose is an essentially linear polymer made by a single enzyme, Granule-Bound Starch Synthase (GBSS)[3]. Observed variations in the number, size, and morphology of starch granules are highly species specific[4,5] and the morphological characteristics of starch in archeological remains are used as identifiers of crop species used by early civilizations[6,7]. However, despite the global importance of starch for human civilizations, the genetic and cell-biological basis for the structural and morphological diversity of starch are largely unknown.

In leaves, starch is formed in chloroplasts during the day from photo-assimilated $CO_2$ and degraded to support metabolism at night, when photosynthesis cannot occur. Arabidopsis chloroplasts were reported to produce 5–7 lenticular starch granules on average[8]. The starch biosynthetic enzymes are well known[9]: α-1,4-linked glucan chains are elongated by a set of five starch synthases (SS: E.C. 2.4.1.21). The α-1,6-branches are introduced by two branching enzymes (BE: E.C. 2.4.1.18) and the structure is finally tailored by isoamylase-type debranching enzymes (ISA: E.C. 3.2.1.68) to promote its crystallization into a lamellar structure with a 9–10-nm periodicity. The capacity to form insoluble, semi-crystalline starch-like granules was recently engineered into the non-starch synthesizing yeast *Saccharomyces cerevisiae* through the introduction of the Arabidopsis enzymes. This synthetic biology approach demonstrated that the core components of starch biosynthetic apparatus are identified[10].

The starch synthase isoform STARCH SYNTHASE 4 (SS4) is known to play a key role in starch granule initiation, with *ss4* mutant chloroplasts exhibiting a strongly diminished number of granules[11–13]. Furthermore, *ss4* mutant granules are aberrant in morphology, i.e., more spherical than lenticular. The abilities of SS4 to both promote granule initiation and control their growth depend on its C-terminal glucosyltransferase domain and on its non-enzymatic N-terminus, respectively[14]. However, the molecular mechanisms underpinning SS4 function are not understood. Recent genetic studies have identified additional proteins involved in granule initiation and growth[15,16]. These include PROTEIN TARGETING TO STARCH (PTST) family proteins, which are proposed to deliver glucan substrates to SS4[17], SS5, a non-enzymatic homolog of SS4[18], and two coiled-coil domain proteins MAR-BINDING FILAMENT-LIKE PROTEIN 1 (MFP1) and MYOSIN-RELATED CHLOROPLASTIC PROTEIN (MRC) that bind to and help localize PTST2 and/or SS4[19,20]. The precise molecular functions of these proteins are also unclear due to the lack of knowledge about the actual process of starch granule initiation.

To discover how starch granule initiation occurs and to track subsequent granule development, we employed a suite of complementary microscopic methods. Serial block face scanning electron microscopy (SBF-SEM) allowed us to visualize the initiation processes in 3D and quantify changes in granule numbers. The additional use of carbon isotope labeling and quantitative nanoscale secondary ion mass spectrometry (Nano-SIMS), in combination with molecular genetic approaches, enabled us to understand the role of granule initials and follow granule expansion patterns during the day. These findings provide a new level of mechanistic insight into the cell biology of starch biosynthesis.

## Results

**Starch granule quantification during the day**. We used SBF-SEM to image mesophyll cell volumes from leaves of wild-type plants (Supplementary Movies 1–4). Leaves sampled at the end of a 12-h day had $7.0 \pm 0.7$ mg g$^{-1}$ fresh weight (FW) starch (SD, $n = 4$, measured enzymatically) and their chloroplasts contained $12.8 \pm 2.3$ (SD, $n = 8$) lenticular granules (determined by manually segmenting the SBF-SEM image stacks). Leaves sampled at the end of the night had $0.24 \pm 0.03$ mg g$^{-1}$ FW (SD, $n = 4$) starch and $4.4 \pm 1.0$ (SD, $n = 5$) granules per chloroplasts, which appeared as small discs, thinned at the poles (Fig. 1a). Thus, only 3% of the mass of starch accumulated in the leaf during the day remained at the end of the following night, but the cores of 35% of the granules were preserved. These cores presumably serve as substrates for renewed starch synthesis during the next day, in parallel with the initiation of new starch granules. To visualize starch granule development, we sampled leaf tissue after 15 min, 30 min, and 8 h light. There were no striking changes after 15 or 30 min (Fig. 1b–c, e), with granule shape similar to that at the end of the night. After 8 h light, chloroplasts contained more granules with one or occasionally more granules in a given stromal pocket (the stroma is continuous within the chloroplast but we define stromal pockets as enlarged regions containing one or more starch granules in close proximity to each other but not separated by thylakoid membranes; Fig. 1d). However, in this experiment, granules newly initiated in the light could not be distinguished from pre-existing granules.

To observe newly initiated granules, chloroplasts were further de-starched by exposing plants to a 4-h extension of the night. We repeated the SBF-SEM analysis, sampling after the long night and 15 min, 30 min, and 8 h into the day (Fig. 1f–j). Image stacks were manually segmented to delineate the starch-containing pockets and their respective starch granules at different time points. The image segmentation yielded the number of starch granules per chloroplast, the number of pockets, and the number of granules per pocket (cluster size) (Fig. 2). As expected, chloroplasts contained no starch granules after 16-h darkness, but each chloroplast had several regions where the stroma was less electron-dense and floccular in appearance (Fig. 1f, Supplementary Fig. 1 arrowheads; Supplementary Movie 4). When exposed to light for 15 min, some of these regions (on average, five per chloroplast) contained starch granule initials (Fig. 1g, j). Occasionally there was a single granule initial, but usually a cluster was observed (with an average of 4.1 initials, with one pocket exceptionally containing 28 initials: Fig. 2a–d; Supplementary Movie 1). After 30 min light, the number of pockets was unchanged, but the diameter of the granules had enlarged, many had an irregular appearance, and the average number of granule initials per cluster had decreased to 3.1 (Fig. 1h; Fig. 2a–d; Supplementary Movie 2). After 8 h light, chloroplasts contained regular lenticular (oblate spheroid) granules, occupying more pockets than earlier in the day (11 on average). At this time there were more granules per chloroplast than after 30 min (averages of 20 and 16, respectively), but the cluster size had decreased further (on average, 1.9, with typically just one or two; Fig. 1i; Fig. 2a–d; Supplementary Movie 3). Interestingly, when there was more than one granule per stromal space, the abutting surfaces were flat (e.g., Fig. 1i, arrowhead). We computed the total surface area and starch volume for each chloroplast and derived the surface-to-volume ratio. As expected, surface area and volume

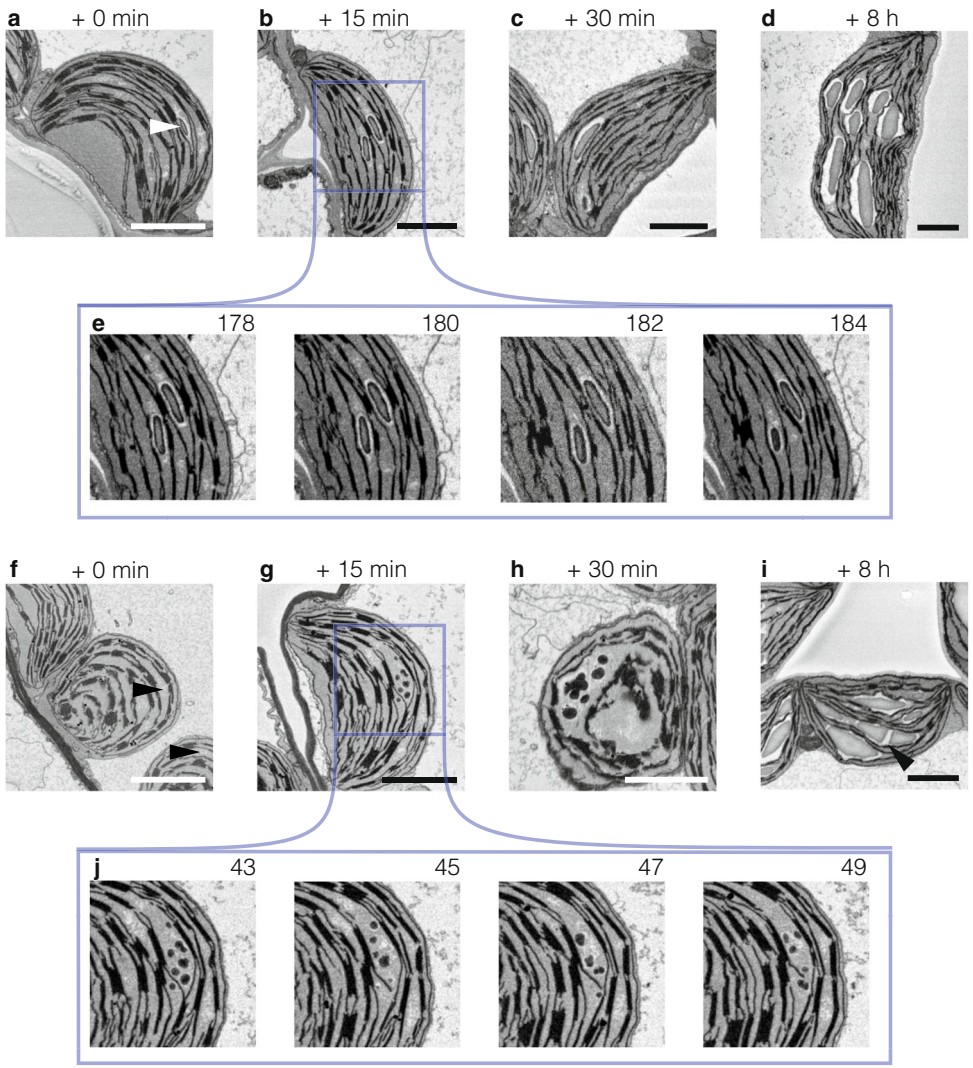

**Fig. 1 De-starched chloroplasts exhibit parallel initiation of starch granules shortly after dawn.** Chloroplast sections from an SBF-SEM stack (with 50-nm Z-resolution, inverted SEM images) sampled from the mesophyll of fully expanded leaves of 35-day old Arabidopsis plants grown in a 12 h:12 h diel regime. Starch granules appear as dark-gray disks between thylakoid membranes. **a–d** Representative chloroplast sections from plants harvested (**a**) at the end of a regular 12-h night, with a remaining starch granule (white arrowhead), (**b**) after 15 min light, (**c**) after 30 min light, and (**d**) after 8 h light. **e** Series of images (with 100-nm steps in the Z axis) from the sample stack in **b**. **f–i** Representative chloroplast sections from plants exposed to a prolonged night and harvested (**f**) at the end of the 4-h night extension; arrowheads indicate stromal spaces with a floccular appearance (**g**) after 15 min light, (**h**) after 30 min light, and (**i**) after 8 h light; arrowhead indicates parallel surfaces of abutting granules. **j** Serial sections (with 100-nm steps in the Z axis) from the sample stack in **g**. Note the numerous starch initials. Scale bars: 2 µm. Further data are given in Supplementary Fig. 1 and Supplementary Movies 1–4.

both increased at the later time points, while the surface area-to-volume ratio declined (Fig. 2e).

**Coalescence and anisotropic growth of starch granules.** The fact that granule number per chloroplast was highest after 15 min light (on average, >20) and decreased by ~50% after 30 min light without a change in the number of starch-containing pockets suggests that either some starch granule initials are degraded again, or granule initials coalesced as starch synthesis proceeded. The irregular appearance of granules after 30 min is suggestive of the latter, implying that granules can have multiple initiations. Furthermore, the fact that after 8 h light, the numbers of both starch granules and starch-containing pockets increased suggests that new granules continued to be initiated during the day.

To determine whether starch granule initials do indeed coalesce during the day and understand how they grow, we performed $^{13}CO_2$ stable isotope labeling of illuminated plants, followed by transmission electron microscopy (TEM) and subsequent high resolution, quantitative isotope mapping of the imaged samples using NanoSIMS[21]. After de-starching chloroplasts by extending the night to 16 h, a pulse of $^{13}CO_2$ air was given at the start of the day to label newly formed starch during the first 15 min of photosynthesis. Leaf samples harvested immediately after the pulse, or after a 15-min, 45-min, or 4-h chase in normal air, were chemically fixed and embedded in resin for TEM analysis. During fixation and dehydration, most soluble $^{13}C$-labeled compounds are lost from the sample, while $^{13}C$-labeled cellular structures, including starch, are preserved. All of the starch initials clustering in pockets were labeled after the pulse of $^{13}CO_2$, showing that they were newly synthesized (Fig. 3a). After subsequent 15-min and 45-min chases, we observed unambiguous coalescence events where granules

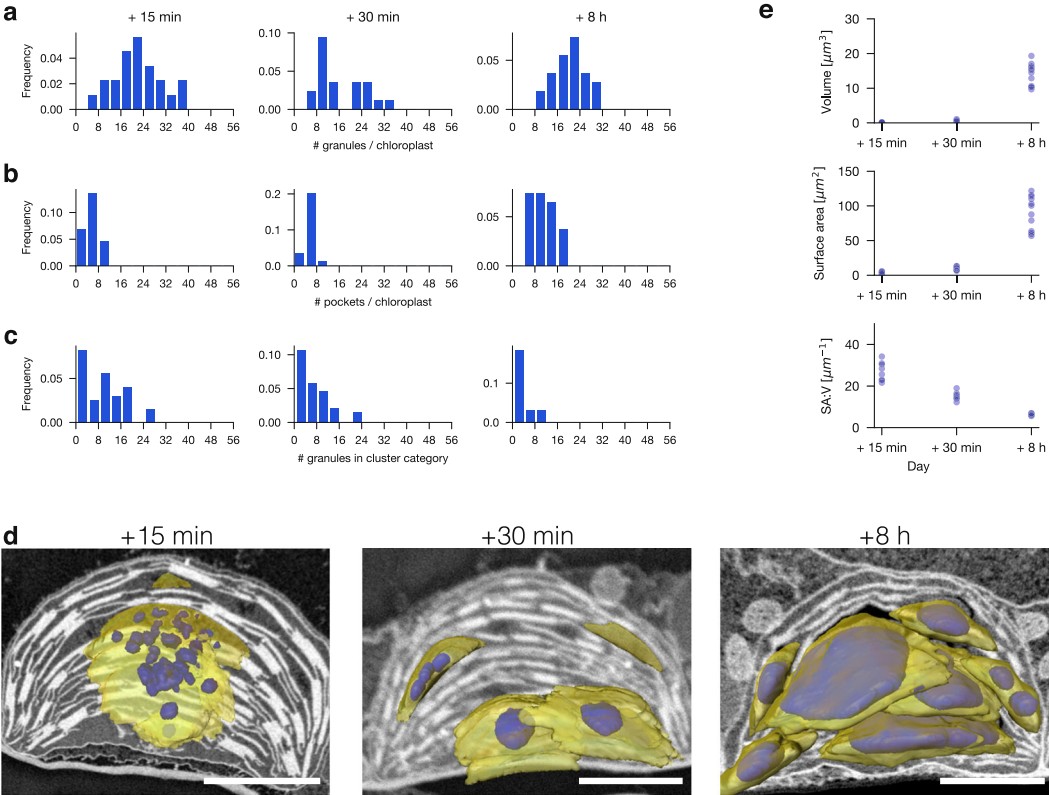

**Fig. 2 Imaging of entire chloroplasts to determine the exact number of starch granules.** Tomographic reconstruction from SBF-SEM image stacks of entire chloroplasts from plants sampled after a prolonged night and after 15 min, 30 min, and 8 h light. **a** The number of granules per chloroplast. **b** The number of starch-containing pockets per chloroplast. **c** Number of granules in each cluster category. Data in **a**–**c** are expressed as frequency distributions (bins of 4, including 0 in **a** and **b**, excluding 0 in **c**), where the height of the bar represents the frequencies within each bin. For each time point, two plants from two independent experiments were examined. In total, 22, 21, and 27 chloroplasts were examined for the 15-min, 30-min, and 8-h time points, respectively. **d** 3D-renderings of representative chloroplasts (non-inverted SEM images) with the pockets (yellow) and their starch granules (violet). Scale bars: 2 μm. **e** The volume (top), surface area (middle), and the surface area-to-volume ratio (SA:V; bottom) for the total starch in each chloroplast. In total, 8, 8, and 12 chloroplasts were examined for the 15-min, 30-min, and 8-h time points, respectively, to determine these values.

forming around labeled starch initials had started to fuse (Fig. 3b and c). After the 4-h chase, starch granules could be observed with more than one distinct region of $^{13}C$-enrichment embedded within them (Fig. 3d). For further examples, see Supplementary Fig. 2a–d. Thus, newly initiated, labeled granules continue to grow and fuse using unlabeled photoassimilates produced during the chase. Interestingly, in harvests performed after longer chase periods (Fig. 3d), some granules appear unlabeled, which is presumably because the labeled center is not in the plane of the ultrathin section, or because the granule initiated after the 15-min pulse.

Next, we examined more closely the pattern of granule growth after these initiation and coalescence events. First, plants treated with an extended night (16 h) were allowed to photosynthesize for 45 min in the light, then labeled for 15 min with $^{13}CO_2$. In leaves sampled immediately afterwards, all granules were labeled, predominantly at the surface with much less enrichment in the existing granule core. The pattern of surface labeling was non-uniform, with much more label incorporated onto the margins of the granules than onto the faces, showing that they were expanding rapidly equatorially, and less so at their poles (Fig. 4a; Supplementary Fig. 2e). This indicates that an expansion pattern is established early in the day and dictates the characteristic lenticular granule shape. In a second experiment, plants treated with a normal 12-h night were labeled with $^{13}CO_2$ at midday for 1 h (to achieve sufficient labeling over the increased granule surface area). As for the first experiment, most label was incorporated non-uniformly onto the granule surface, with stronger

enrichment along the margins than on the faces (Fig. 4b; Supplementary Fig. 2f). Interestingly, where granules abutted, label incorporation was also observed on the flattened surfaces. We quantified the enrichment on the granule surface contour and mapped it to the unit circle, starting at a point furthest from margins, i.e., closest to the barycenter of the granule, to obtain enrichment profiles. Aligning the enrichment profiles revealed that regions of rapid growth coincided with the granule margins and were characterized by several-fold higher $^{13}C$-label than the slow-growing granule faces (Fig. 4c). However, not all granules showed rapid growth equally on both margins. Furthermore, rapid growth was less evident on the flattened surfaces of abutting granules.

**Disrupted starch biosynthesis patterns in Arabidopsis mutants.** The availability of Arabidopsis mutants was used to determine which factors control the patterns of starch biosynthesis. First, we analyzed the *ss4* mutant, which is defective in the initiation and growth of granules[11]. The chloroplasts of *ss4* have few large, rounded starch granules (opposed to numerous, lenticular granules in the wild type). SBF-SEM revealed that, after a 4-h night extension, some *ss4* chloroplasts still contained a granule, unlike the wild type (Supplementary Fig. 3a). After 15 or 30 min of light, a few small granules were present in addition to the larger granules inherited from the day before (Supplementary Fig. 3b and c). These observations, together with the presence of both large and small granules after 8 h (Supplementary Fig. 3d) suggest

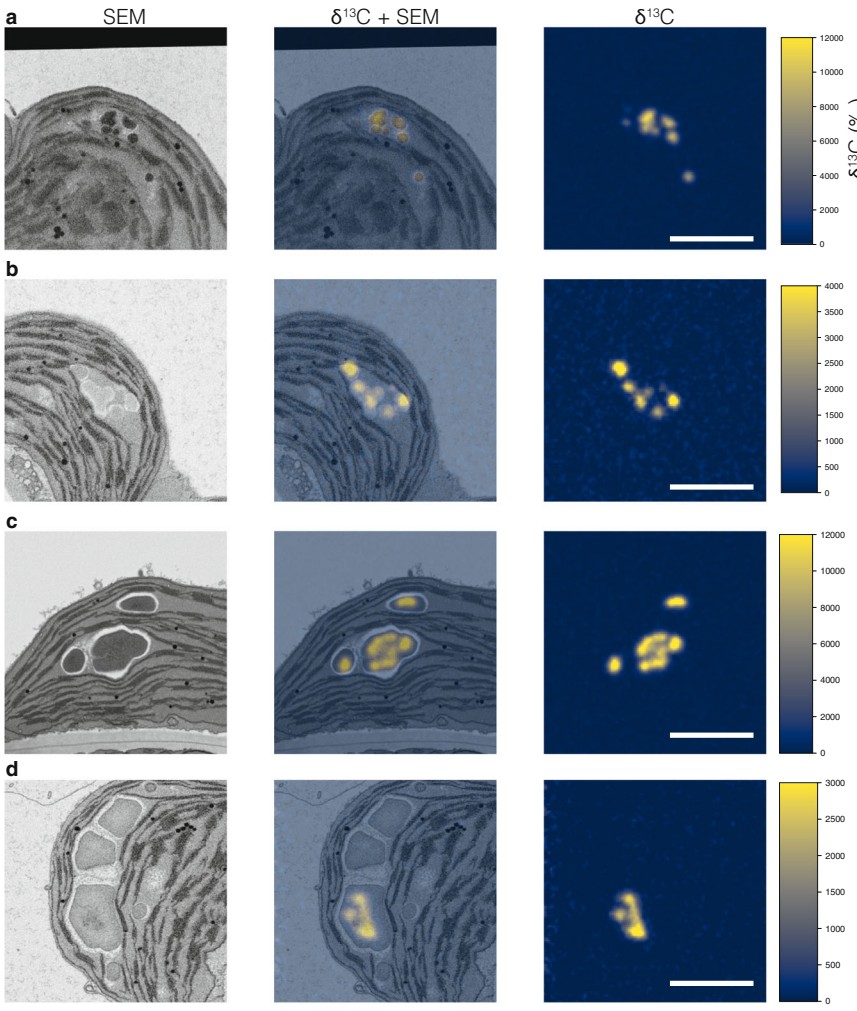

**Fig. 3 Coalescence of starch granule initials tracked with stable isotope labeling, EM, and NanoSIMS imaging.** Plants subjected to a 4-h night extension were labeled with a pulse of $^{13}CO_2$ for 15 min in the light, then harvested immediately (**a**), or after a chase of 15 min (**b**), 45 min (**c**), or 4 h (**d**) in normal air in the light. In each case, samples were fixed and embedded for EM and NanoSIMS imaging. The electron micrograph (left, inverted SEM image) and the enrichment map (right), and their overlay (center) are shown. The $^{13}C$-enrichment is reported as per mille. Note the multiple labeled initials in each stromal space, with an increasing degree of coalescence during the chase in normal air. Scale bar: 2 μm.

that initiation occurs, albeit at a much-reduced rate. Interestingly, we never observed clusters of granule initials similar to those seen in the wild type (Supplementary Fig. 3e, Supplementary Fig. 4). To study the pattern of granule initiation and growth, we subjected *ss4* plants to an extended night and pulsed them with $^{13}CO_2$ for 15 min in the light, sampling leaves after a 4-h chase (Supplementary Fig. 5). Large granules contained a central ring of $^{13}C$-label, indicating that these granules had remained after the extended night and were serving as surfaces for starch re-growth. Interestingly, other granules, sometimes within the same pocket, appeared centrally labeled, consistent with the initiation of new granules, even with pre-existing granules present. We found no evidence of granules containing multiple initials, consistent with the absence of clusters in *ss4*.

We further studied the pattern of granule growth by labeling *ss4* plants with $^{13}CO_2$ for 1 h in the middle of the day (after a 12-h night), followed by immediate sampling. The almost uniform, isotropic labeling of the granule surface in *ss4* suggests that SS4 is required for normal anisotropic granule growth (Fig. 5a, Supplementary Fig. 6a) and explains the mutant's rounded granule morphology. The granule initiation phenotype of *ss4* can be rescued by the transgenic expression of a self-glycosylating

bacterial glycogen synthase[14,22]. Similar $^{13}C$-labeling experiments performed on these transgenic plants revealed that each chloroplast contained numerous round granules that were uniformly labeled. This confirmed that initiation was increased, but that granule growth remained isotropic (Fig. 5b, Supplementary Fig. 6b). When appended to the bacterial glycogen synthase, the non-enzymatic N-terminal domain of the SS4 protein influenced its sub-chloroplastic localization[14,23] and rescued both the granule number and granule morphology phenotypes of the *ss4* mutant[14]. Labeling experiments on these transgenic plants revealed that each chloroplast contained numerous lenticular granules, with label concentrated on their equatorial regions, similar to the pattern in the wild type (Fig. 5c, Supplementary Fig. 6c). Thus, correct localization of glucan initiating activity by the N-terminus of SS4 is sufficient to guide the overall starch biosynthesis process. This is remarkable because starch is produced by a suite of biosynthetic enzymes, including but not limited to SS4. In the absence of SS4, these enzymes appear to operate uniformly on the available granule surface.

Finally, we analyzed the synthesis of amylose - the minor starch component formed by GBSS. When labeled for 1 h in the

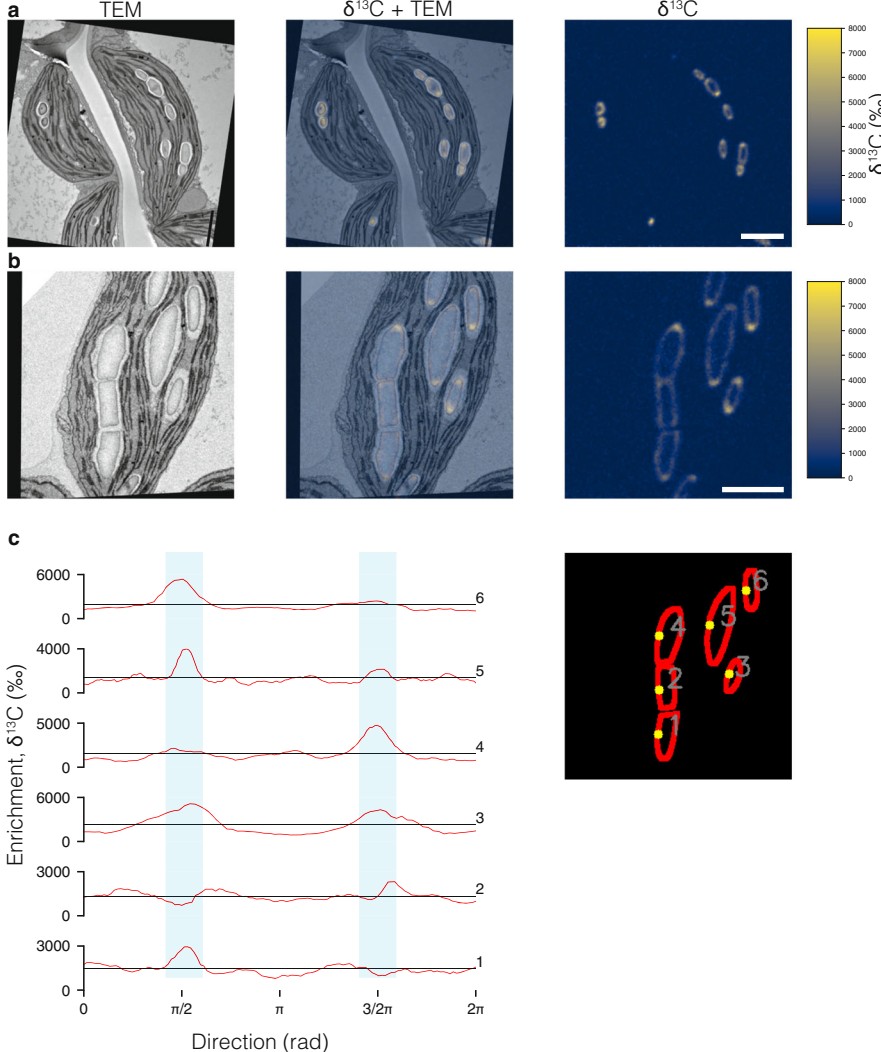

**Fig. 4 Anisotropic starch granule expansion revealed by stable isotope labeling, EM, and NanoSIMS imaging. a** Plants subjected to a 4-h night extension photosynthesized for 45 min in normal air, were labeled with a pulse of $^{13}CO_2$ for 15 min, and then harvested immediately for EM and NanoSIMS imaging, as in Fig. 3. **b** Plants subjected to a normal night were labeled with a pulse of $^{13}CO_2$ for 1 h at midday then harvested immediately. Scale bars: 2 μm. **c** Quantification of $^{13}C$-enrichment on the surface of six granules depicted in **b**. The probing regions (depicted in red in the inset, right) were defined manually with masks over the $\delta^{13}C$ map. Individual circular profiles were extracted anticlockwise from the starting point (yellow). Light blue segments highlight the parts of the profiles corresponding to the granule margins, where most enrichment is observed. Horizontal lines show the mean enrichment for each profile.

middle of the day (after a 12-h night), granules from the amylose-free *gbss* mutant (which lacks the GBSS protein) displayed a similar surface-labeling pattern as in the wild type (Fig. 6a and b). However, when granules were manually segmented into surface and internal core regions, there was less internal enrichment in the *gbss* granules, compared with the wild type (Fig. 6c). In contrast, granule surface enrichment was slightly higher in *gbss* than in the wild type (Fig. 6c). These data show unambiguously that GBSS-mediated amylose synthesis occurs inside the starch granule matrix in vivo.

## Discussion
Understanding the in vivo biosynthesis of biomacromolecules requires genetic, biochemical, and cell-biological insight. We have brought together a new combination of experimental methods and analytical techniques to bridge these disciplines in the study of starch—arguably the most important storage biopolymer produced in plants and a vital agricultural product. As with many

biopolymers, starch granule formation is brought about by the concerted actions of numerous enzymes, the coordination of which is evidently genetically controlled, but poorly understood at the cell-biological level. The study of granule initiation in Arabidopsis provides us with a tractable genetic system with which to identify the genes and proteins controlling these processes[11,14,15,19,20,24,25]. By combining high-resolution spatio-temporal imaging techniques with genetics, we bridge these disciplines to address previously intractable questions about biopolymer formation in vivo. Collectively, the parallel initiation, followed by fusion and localized starch accumulation, challenge the classical view of granule development and illustrates its dependence on the emerging network of proteins that includes both enzymes and scaffolding factors.

There is strong evidence from this work (Fig. 1) and previous studies[19] that starch granule initiation in chloroplasts occurs at defined sites in-between the thylakoid membranes, i.e., the stromal pockets we describe here. Some proteins implicated in initiation are themselves partly (SS4, PTST2) or wholly (MFP1)

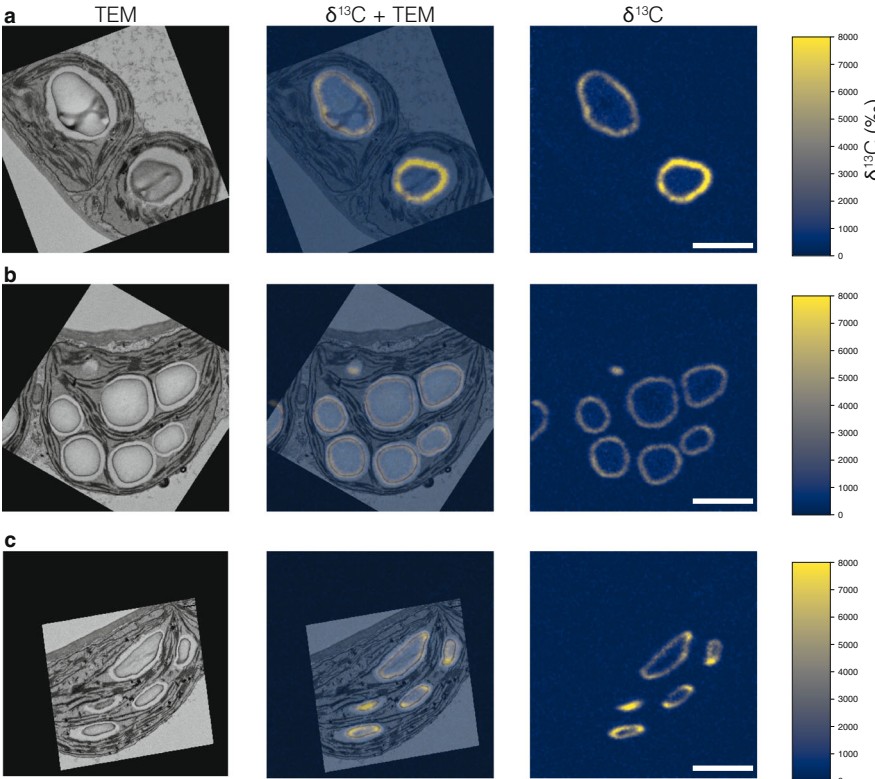

**Fig. 5 Starch granule initiation and growth are controlled by STARCH SYNTHASE 4.** Plants subjected to a normal night were labeled with a pulse of $^{13}CO_2$ for 1 h at midday then harvested immediately for EM and NanoSIMS imaging, as in Fig. 3. **a** Chloroplasts from the *ss4* mutant. **b** Chloroplasts from the *ss4* mutant expressing a self-glycosylating glycogen synthase (GS) from *A. tumefaciens*. **c** Chloroplasts expressing the GS fused with the N-terminal of SS4. Scale bars: 2 μm. For additional examples, see supplementary Fig. 2g–i. For analysis of the patterns of $^{13}C$-enrichment, see Supplementary Fig. 6.

thylakoid-associated and localize to chloroplast subdomains[19,23,25]. Further, our SBF-SEM analysis of de-starched plants, i.e., after 16-h nights, revealed thylakoid-bounded regions differing in appearance to the rest of the stroma. We propose that these are the subdomains where the granule initiation proteins accumulate, thereby defining stromal pockets where granule initials will form. This is a hypothesis we are testing with other correlative microscopy methods. Previous models of starch granule biosynthesis generally imply a single point of initiation—the hilum—and suggest that when multiple initiations occur, compound starch granules are formed, with each with its own hilum (as seen in the endosperms of some cereals[26,27]). Our study unambiguously demonstrates that single granules can arise from multiple parallel initiations that subsequently coalesce. In chloroplasts, this pattern of multiple initiations may be functionally important in allowing the rapid establishment of sufficient granule surface area, which is key to the high rate of starch deposition that accompanies photosynthesis. It will be important to determine next whether similar granule initiation patterns occur in the plastids of other plant storage organs, such as in amyloplasts of seeds and tubers, and whether this process influences starch yields.

Granule coalescence events raise fundamental, unanswered questions. The arrays of parallel glucan chains in amylopectin molecules are thought to be radially arranged in starch, such that the 9–10-nm semi-crystalline lamellar repeats are concentric, giving granules their characteristic birefringence when viewed under polarized light[2,28]. It is unclear if these semi-crystalline structures are already formed in the observed granule initials. If semi-crystalline lamellae are present, they are unlikely to be organized in a concentric way considering that, at ~50 nm in diameter, the initials are not much larger than the lamellae themselves. Rather, each initial may be composed of a stack of a few, similarly oriented lamellae. We speculate that, for adjacent granule initials to coalesce, the orientations of the constituent polymers may need to be similar (e.g., parallel), and that initials with opposing polymer orientations (e.g., antiparallel) might not fuse. Indeed, not all adjacent granules in the same pockets fuse, despite the fact that our NanoSIMS imaging revealed deposition of new material onto their flat, abutting surfaces (Fig. 4a). Alternatively, it is possible that both the initials and larger granules can fuse regardless of the orientations of their constituent polymers but that, after a certain stage of granule expansion, fusion is prevented by the protein factors that establish the pattern of anisotropic growth (Fig. 4b).

The anisotropic deposition of newly synthesized material dictates the way starch granules grow in Arabidopsis chloroplasts into flattened discoids that fill the pockets and lie parallel to the thylakoids. For this to occur, the activities of a suite of enzymes (starch synthases, branching enzymes, and debranching enzymes, and/or the substrate generating enzyme, ADPglucose pyrophosphorylase) must be concentrated at the equatorial regions of the granule. In other starch synthesizing tissues, starch granules also take on very specific morphologies. By analogy with the observations made here, we suggest this also happens via the nonrandom patterns of deposition of new material. Remarkably, our data show that SS4 is critically required for growth anisotropy in Arabidopsis; when missing, newly synthesized starch is deposited almost uniformly onto the granule surface, explaining why the starch granules in *ss4* mutants are near-spherical. We conclude that the presence of this protein—specifically its N-terminal domain required for the protein's correct localization coupled to

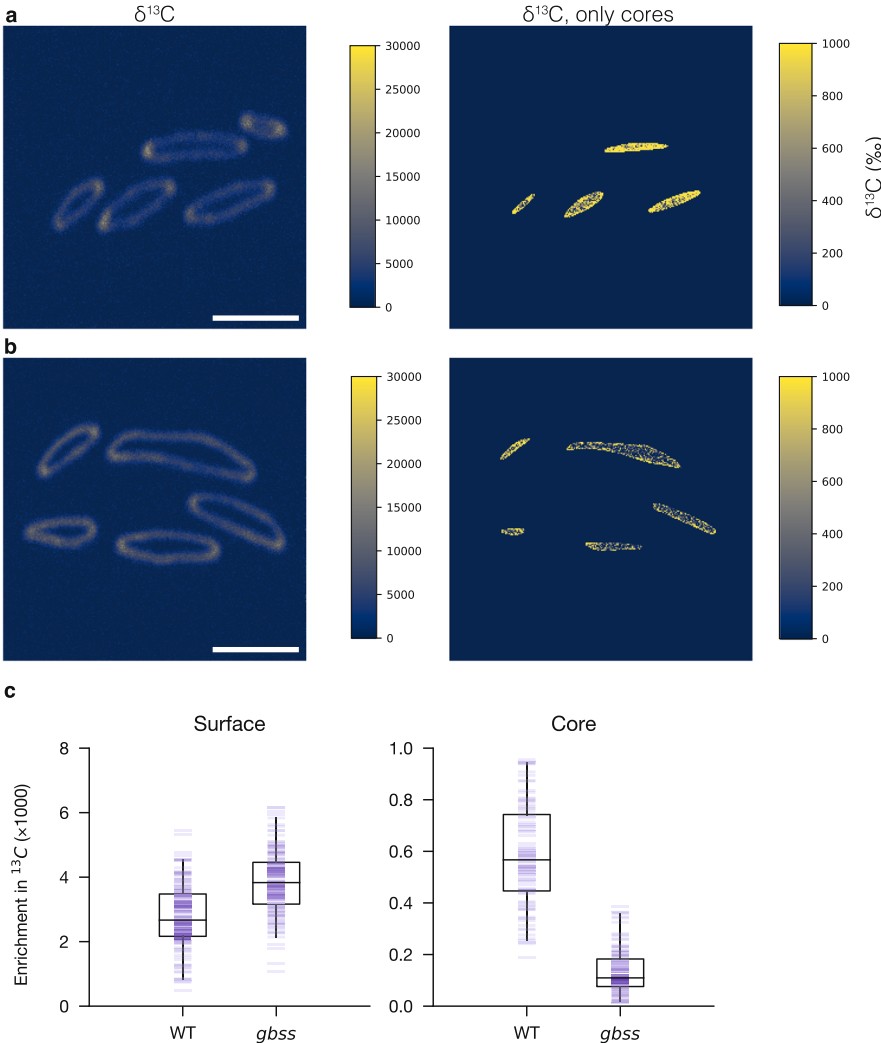

**Fig. 6 GRANULE-BOUND STARCH SYNTHASE synthesizes amylose in the granule cores.** Plants subjected to a normal night were labeled with a pulse of $^{13}CO_2$ for 1 h at midday then harvested immediately for EM (not shown) and NanoSIMS imaging. Scale bars: 2 μm. **a** $^{13}$C-enrichment map (left) of starch granules within wild-type chloroplasts. Manually masking the granule surfaces revealed enrichment specifically in the cores (right; note different $^{13}$C-enrichment scales). **b** $^{13}$C-enrichment map of starch granules (left) and their cores (right) within chloroplasts of the *gbss* mutant. **c** Quantification of the $^{13}$C-enrichment of granule cores and surfaces overlaid with boxplots, where the minima, maxima of the whiskers represent 2.5% and 97.5%, the bounds of box, 25% and 75%, and the center denotes 50% of the posterior. $N = 259$ and 269 granules for the wild type and *gbss* mutant, respectively, in each case from three biological replicates.

glucan synthase activity—is crucial for concentrating the activities of the other starch biosynthetic enzymes, including SS1–SS3, BE2 and BE3, and ISA1/2. We propose that a combination of protein-glucan, protein–protein, and protein–lipid interactions underpins this phenomenon. SS4 is known to interact with other proteins, i.e., the fibrillins FBN1a and FBN1b, PTST2, and MRC[19,20,24,25], which themselves have been demonstrated or proposed to have binding capabilities. The PTST-family proteins can bind starch and malto-oligosaccharides, and, like the putative structural proteins MRC and MFP1, contain predicted coiled-coil motifs that could mediate protein–protein interactions[19,20,24]. Further, MFP1 is known to be a thylakoid membrane-associated protein[19,29]. Hence, SS4 may serve as a vital link in this inter-action network. Other starch biosynthetic enzymes also have predicted glucan- and/or protein-interaction domains[10]. Indeed, in cereal species, multi-enzyme complexes containing other starch synthase isoforms and branching enzymes have been shown to form[30,31]. In addition to coordinating their activities,

the formation of multi-enzyme complexes could facilitate their collective sub-plastidial localization.

It is possible that the sites of granule synthesis in stromal pockets may be examples of liquid–liquid phase-separation brought about by many low-affinity protein–protein and protein-glucan interactions. A number of the aforementioned proteins have amino-acid regions predicted to be intrinsically disordered—a common feature in proteins capable of condensing into phase-separated organelles not enveloped by membranes. Starch biosynthetic enzyme activities may also be significantly increased by high local concentrations of glucan substrates within such condensates (while the actions of degradative enzymes might be simultaneously excluded). This type of enzyme stimu-lation has been demonstrated for GBSS, the activity of which shows biphasic kinetics as soluble glucan substrate concentrations breached a threshold, above which viscosity and turbidity increased, signifying the onset of phase transition. This, together with other observations, prompted the hypothesis that GBSS is

active within the granule matrix[32–34], which we demonstrate to be correct through isotope labeling and NanoSIMS imaging.

In conclusion, the power of combining genetics with innovative imaging techniques to capture temporal and spatial information with high resolution has allowed us to answer previously intractable questions about starch biosynthesis. This approach, extended to the study of other important biopolymers in vivo, could enable similar breakthroughs in those fields.

## Methods

**Plant material and growth conditions**. All experiments were carried out with *Arabidopsis thaliana* L., ecotype Col-0, grown on individual pots of fertilized bedding compost (Klasmann-Deilmann Substrate 2) for 35 days in controlled environment cabinets (Percival Scientific, Perry, IA, USA) with a 12-h light (150 μmoles photons m$^{-2}$ s$^{-1}$)/12-h dark regime, with regular watering. The relative humidity was 60% and the temperature was 20 °C. Plants defective in SS4 (At4g18240) and their transformed lines expressing *Agrobacterium tumefaciens* glycogen synthase, or glycogen synthase linked to the N-terminus of SS4, are those that were described previously[14]. Plants defective in GBSS (At1g32900) were also described previously[35].

**Stable isotope labeling**. For $^{13}CO_2$ labeling (either at dawn, or after a 4-h night extension for further de-starching), plants were transferred to an airtight chamber within the growth cabinet. Air was passed through the chamber in which $CO_2$ was substituted with 380 ppm $^{13}CO_2$ (Pangas, Dagmersellen, Switzerland). Labeling to investigate granule initiation was started 5 min before dawn and continued for the first 15 min of the light period. Labeling to investigate granule growth was done for 60 min at midday. Plants were harvested immediately after the $^{13}CO_2$ pulse or transferred to the growth cabinet for a defined chase period.

**Serial block face scanning electron microscopy**. Samples were taken from the upper quadrant of the lamina of unshaded leaf #6, between the major vein and the leaf margin, for all imaging experiments. Each fixation and staining step was followed by three washing steps with the corresponding buffer. Leaf samples were fixed for 6 h at 20 °C, 200 mbar in fixation solution (2.5% [v/v] glutaraldehyde, 2% [v/v] formaldehyde in 0.1 M sodium cacodylate, pH 7.4). Samples were then stained in a medium containing 1.5% (w/v) potassium ferrocyanide, 2% (w/v) osmium tetroxide, 4 mM calcium chloride for 1 h on ice, followed by an incubation in pre-filtered 1% (w/v) thiocarbohydrazide solution (20 min, 20 °C) then in 2% (w/v) osmium tetroxide (in 0.1 M sodium cacodylate, pH 7.4, 30 min, 20 °C). Samples were then stained with 1% (w/v) uranyl acetate for 12 h at 4 °C and then with Walton's lead aspartate for 30 min at 60 °C (0.66% [w/v] lead nitrate in 0.4% [w/v] aspartic acid, pH 5.5). Samples were dehydrated with a six-step ethanol series (50%, 60%, 70%, 80%, 98%, 4 × 100%), followed by 100% acetone, each for 20 min at 4 °C. Samples were infiltrated in epoxy resin (Durcupan™ ACM, Sigma-Aldrich, Buchs, SG, Switzerland) with increasing concentrations (25%, 50%, 75%, 3 × 100%) in acetone. The infiltrated samples were cured in molds for 48 h at 60 °C[36]. The blocks were trimmed, glued on pins, and sectioned/scanned ca. In all, 500 times using a scanning electron microscope (FEI Quanta 250, Thermo Fisher Scientific) at 1.8–2 kV with an integrated 3View stage and a back-scattered electron detector (Gatan, Pleasanton, CA, USA). The structural features were marked out manually with Amira (ThermoFisher, Waltham MA, USA).

**Sample preparation for EM-NanoSIMS imaging**. When TEM was used for section imaging, leaf samples were fixed with fixation solution (2.5% [v/v] glutaraldehyde, 2% [w/v] formaldehyde in 0.1 M sodium cacodylate, pH 7.4), stained with 1% (w/v) osmium tetroxide, dehydrated with a six-step ethanol series, infiltrated with a four-step (25%, 50%, 75%, 3 × 100%) epoxy resin series in ethanol (Spurr Low Viscosity Embedding Kit, Polysciences Inc, Warrington, PA, USA) and cured for 48 h at 60 °C. All steps except the last were microwave-assisted using a BioWave Pro+ (Ted Pella, Redding, CA, USA), with 40 s/250 W settings for dehydration steps and 5 min/200 W/$T_{max}$ = 22 °C settings for embedding[18]. Ultrathin sections (70 nm) were transferred onto formvar-coated copper grids (EMS, Hatfield, PA, USA), post-stained with uranyl acetate (10 min) and with lead citrate (10 min). Images were acquired with a transmission electron microscope (JEOL-1400 Plus, JEOL Inc. Peabody, MA, USA) at 120 kV.

When SEM was used for section imaging, samples were processed as for TEM, but osmium staining prior to dehydration and embedding was based on the OTO method[37], adapted as follows. Fixed samples were stained with 3% (w/v) potassium ferrocyanide (II) in 0.1 M sodium cacodylate, pH 7.4, with an equal volume of 4% (w/v) osmium tetroxide solution. Next, samples were incubated with pre-filtered 1% (w/v) thiocarbohydrazide solution and finally stained with 2% (w/v) osmium tetroxide solution. After embedding, 200-nm-thick sections were cut, collected on hydrophilised wafers (PELCO easiGlow™ Glow Discharge Cleaning System), and imaged with a GeminiSEM 500 (ZEISS, Oberkochen, Germany) scanning electron microscope (3 kV) using the back-scattered electron detector.

**Nanoscale secondary ion mass spectrometry**. Grids previously imaged with TEM or by SEM (semi-thin samples deposited on silicon wafers) were coated with ca. 15 nm gold. Using a NanoSIMS 50 L instrument[20], pre-sputtered regions of interest (ROI) of 20 × 20 μm$^2$ or 10 × 10 μm$^2$ (256 × 256 pixels) were scanned with a 16 keV primary Cs$^+$ beam, focused to a spot size (Gaussian shape, FWHM) of ~100 nm with a pixel dwell time of 5 ms. For each image, the counts for the secondary ions $^{12}C_2^-$, $^{13}C^{12}C^-$, and $^{14}N^{12}C^-$ were recorded over five rasters (planes) in electron multipliers at a mass resolving power of around 9000 (Cameca definition), enough to resolve the ions of interest from potential interferences in the mass spectrum. For each session, three analyses were done on similarly prepared, unlabeled plant material serving as isotopic standard against which $^{13}C$-enrichments were reported in the δ-notation, according to the equation:

$$\delta^{13}C(\text{‰}) = ((C_{mes}/C_{nat}) - 1) \times 10^3,$$

where C_mes is the measured $^{12}C^{13}C^-/^{12}C_2^-$ ratio of the sample and C_nat is the average $^{12}C^{13}C^-/^{12}C_2^-$ ratio measured in unlabeled samples (isotopic controls).

Using the ImageJ plugin OpenMIMS, ion map files (*.im) were corrected for dead time and aligned using the $^{14}N^{12}C^-$ map, which shared the highest similarity with the corresponding electron micrograph. All further image processing tasks were carried out in Python the ImageJ plugin OpenMIMS. The aligned and accumulated planes of $^{12}C^{12}C^-$ and $^{13}C^{12}C^-$ were used to compute the $^{13}C$-enrichment maps (in δ$^{13}C$) and electron micrographs were then overlaid onto these maps using the $^{14}N^{12}C^-$ map as a reference. Differences observed in the degree of enrichment could result from variation in (i) assimilation rates between plants, (ii) in the incident light and/or $CO_2$ reaching a given chloroplast, and (iii) the amount of glucan present, i.e., the number stromal pockets in which granules can form or existing granule surfaces available for elaboration. To compare the $^{13}C$-enrichment in granule cores between the wild type and the *gbss* mutant, the ion maps were manually segmented into three regions: background, granule surface, and granule core. For each region, the average value of the enrichment along with the standard deviation was calculated.

**Statistical analysis**. For each inferential procedure, we used Stan[38] to draw posterior samples from the probability density of generalized linear models (Gamma likelihood and a normal prior Normal(mu=2, sigma=2)). The parameters were estimated using the Hamiltonian Monte-Carlo NUTS sampler using four chains, each with 16,000 iterations. The last 8000 iterations were used for analysis. The plots were drawn using the Python plot library Matplotlib.

**Reporting summary**. Further information on research design is available in the Nature Research Reporting Summary linked to this article.

## Data availability

Source data are provided with this paper and can be found in the ETH Zurich Research Collection (https://doi.org/10.3929/ethz-b-000508434).

## Code availability

Python scripts used during data analysis are available from GitHub (https://github.com/leoburgy/buergy_ncomms_2021).

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

## Acknowledgements

We thank Miriam Lucas (ScopeM, ETH Zürich) and Louise Jensen (LGB, EPFL) for guidance with electron microscopy, Florent Plane for NanoSIMS technical support, Federico Massini for help with image segmentation, and Andrea Ruckle for help with plant culture. This work was funded by the Swiss National Science Foundation (grants CR32I3_166487 and 31003A_182570) and by ETH Zurich.

## Author contributions

L.B. contributed to the investigation, methodology, formal analysis, visualization, data curation, writing, and the original draft. S. Eicke contributed to the investigation, methodology, and visualization. C.K. and C.J. contributed to the investigation. K.J.L. contributed resources. S. Escrig contributed to the methodology. A.M. contributed to the conceptualization, methodology, writing, review & editing. S.C.Z. contributed to the conceptualization, supervision, writing, original draft, review & editing.

## Competing interests

The authors declare no competing interests.
