## [Peer Review File · Nature Communications]

Coalescence and directed anisotropic growth of starch granule initials in subdomains of *Arabidopsis thaliana* chloroplastsReviewers' Comments:

Reviewer #1:

Remarks to the Author:

This manuscript by Bürgy et al., describes the study of starch granule growth in vivo by the combination of electron tomography, NanoSIMS isotope mapping and the use of wild-type Arabidopsis plants or Arabidopsis mutants specifically altered for the number and morphology of starch granule synthesis (ss4- mutant). Indeed, the growth of the starch granule was tracked after incorporation of ^{13}C in starch during short pulse (15 min to 1h) of $^{13}\text{CO}_2$ provided to illuminated Arabidopsis plants and the subsequent observation of the synthesized starch granules in leaf plastids (where starch synthesis occurs).

Starch granule initiation and growth remain, to date, probably one of the least understood processes of the starch pathway in plants. Initiation and growth condition the number and morphology of the starch granule in plants and consequently impacts the rates of synthesis and degradation of the polysaccharide which are tightly regulated. In this work, this is the synthesis of transitory starch that has been analyzed. Transitory starch is starch that accumulates in photosynthetic tissues during the day and is degraded at night. This is typically the type of starch that could be found in Arabidopsis leaves. In that case starch granule synthesis occurs in the stroma of the plastid between thylakoids. Arabidopsis wild-type starch granules are flat-rounded. According to literature, 5 to 7 granules could be generally observed in the plastids at the end of the day.

The topic of this manuscript as well as the results presented are of prime importance in the field of starch metabolism. The authors show that granule growth in WT Arabidopsis is anisotropic (i.e. in a preferred direction) whereas it is isotropic in the ss4- mutant (i.e. in all directions). The results are quite convincing notably those presented in Fig. 4C showing that the incorporation of carbon occurs preferably at the equatorial regions of the granule in the WT.

They also show that the formation of a starch granule occurs after the coalescence of what they call "granule initials" in WT

I do have some comments or suggestions for the authors:

1) I think that it would be interesting for the reader to see the same diagram (Fig. 4C) applied for the ss4- mutant and the GS and NterSS4-GS expressing lines (Fig. 5).

2) I would appreciate a clear definition of what a "pocket" is in the stroma, where the starch granules are synthesized. To me, the stroma is a continuum inside chloroplasts. In this continuum, thylakoids and grana float without the formation of subsets (pockets?) that would be isolated from others. Reading the manuscript suggests that the stroma is organized in pockets isolated from each other, which is not, I think, real.

3) It is not clear how the starch granules were unquestionably identified during SBF-SEM analysis. Was it done manually or after computational analysis?

4) There are strong differences in signal intensity in some NanoSIMS imaging (Fig 3 for instance). Indeed, the signal observed in Fig. 3B or fig. 3D would have been included in the background signal in Fig. 3A or Fig. 3C. Is there any explanation for that? In addition, could it be possible to produce better quality superimposed images (center of Fig. 3, Fig. 4, Fig. 5) by deleting most of the background of NanoSIMS picture that darkens the picture? This would give a better view of the localization of the observed signal.

5) Fig. 6: a gbss- mutant has been used that shows a reduced incorporation of ^{13}C in the core of the starch granule (Fig. 6B) compared to the wild-type (Fig. 6A). Which kind of gbss- mutant is it? Is the GBSS protein still present within the granule but inactive, or a mutant without GBSS in starch? If the second option is correct, could it be that the reduced signal in the gbss- mutant is related to the lack

of the protein within the granule? I do believe that ^{13}C incorporation from $^{13}\text{CO}_2$ is not limited to starch but could also be found in proteins or any other organic compounds that are synthesized during the light phase.

6) Lines 327-329: the authors suggest that, since granule growth is anisotropic and preferentially observed at the equatorial regions of the granule, synthetic enzymes must be concentrated at the same region. This makes sense; however, it could also be that precursor (ADP-glucose) synthesis and/or concentration occurs preferably at the same region through a still not uncovered mechanism, thus privileging granule growth in specific direction.

7) Lines 338: I understand the reference to fibrillins (FBN1a and FBN1b) that were shown to interact with SS4. However, none of the corresponding mutant display a phenotype that is related to that of the ss4- mutant. Thus, the actual implication of FBNs in the process of starch granule initiation is questioned.

8) A comparative analysis was performed between Arabidopsis wild-type and mutant lines for SS4, which is known to affect the initiation process of starch synthesis *in vivo* by significantly reducing the number of granules per chloroplast and modifying their morphology. I think it would have been worth it to include a line with a phenotype opposite to that of the ss4- mutant. Indeed, the isa1- mutant would have been interesting. In that mutant, starch synthesis is altered but not stopped. Numerous small granules are synthesized in the isa1- mutant that could, somehow, correspond to the "granule initials" suggested by the authors.

9) Lines 310-319. I don't exactly understand the demonstration made here. What would be "opposing orientations of lamellae"? Why the lamellae would be unlikely radially oriented in granules initials? Do the authors have any evidence for that? By the way, if granule formation results from the merging of several "granule initials", starch granules would then consist of several hila. I don't remember that Arabidopsis starch granules have been described containing several hila. Moreover, how does the growth ring would be organized in that case? Starch growth rings are organized around the hilum of the granule as already described elsewhere. This raises the question of how growth rings will organize if several "granule initials" merge to form one bigger granule. One last thing, which is about semantic: to my point of view, the 9-10 nm lamellae are not "radially arranged in starch". This is the molecules inside the 9-10 nm lamellae that are arranged radially (the glucans are more or less oriented parallel to the radius of the starch granule). The lamellae are arranged tangentially to the surface of the granule (i.e. they are placed concentrically from the center (hilum) of the granule).
Christophe D'HULST.

Reviewer #2:

Remarks to the Author:

The manuscript is well structured and presented, and I have a few minor suggestions:
more data is needed for explanation of growing conditions, eg. how the authors have provided that test plants were not exposed to water/nutrient stress?

Also, more information is needed for plant material sampling (eg. leaf from which position) and handling prior analyses

line 425, reference stile is incorrect.

Reviewer #3:

Remarks to the Author:

The current study is firmly based on innovative imaging techniques to provide cell-biological insights into starch granule formation and growth.

By applying SBF-SEM and nanoSIMS to Arabidopsis wt and mutants, it is convincingly shown that multiple starch granules are initiated in stromal cavities, and subsequently these initial structures fuse and grow anisotropically and expand equatorially until reaching their final size and shape. The use of selected starch synthesis mutants, in particular *ss4*, indicates that the non-enzymatic domain of STARCH SYNTHASE 4 is vital for anisotropic growth.

The imaging is conducted in an excellent manner and the results are of very high quality and lend unequivocal support to the conclusions that are put forward. One aspect that could perhaps be made a bit clearer is the time frame of starch granule formation. In figure 3D for example, only one starch grain is labeled, while the others are not. In this context the chase of 4 hours could perhaps be set in proportion to the duration of granule formation. Figure 1F would perhaps benefit from an inset with higher magnification.

RESPONSE TO REVIEWER COMMENTS

Reviewer #1:

RESPONSE: *We very much appreciate the positive and detailed comments from Reviewer 1 and address the specific comments below:*

1) I think that it would be interesting for the reader to see the same diagram (Fig. 4C) applied for the ss4- mutant and the GS and NterSS4-GS expressing lines (Fig. 5).

RESPONSE: *We agree and provide these plots as a new Supplemental Fig. S6. The plots reinforce the conclusions of our manuscript – that there is a regular pattern of anisotropic growth, which is lost or becomes irregular in the absence of STARCH SYNTHASE 4, but regained in the NSS4-GS lines.*

2) I would appreciate a clear definition of what a “pocket” is in the stroma, where the starch granules are synthesized. To me, the stroma is a continuum inside chloroplasts. In this continuum, thylakoids and grana float without the formation of subsets (pockets?) that would be isolated from others. Reading the manuscript suggests that the stroma is organized in pockets isolated from each other, which is not, I think, real.

RESPONSE: *we understand the point of the reviewer and have tried to define better in the text what we mean by a ‘stromal pocket’, i.e. a defined volume of the stroma in which we observed numerous starch granule initials forming in close proximity (Lines 99-101). We also altered the discussion (Lines 305-311) to note that if our speculation about phase separation events is correct, this could result in the formation of real membrane-less subdomains within the stroma (Line 368).*

3) It is not clear how the starch granules were unquestionably identified during SBF-SEM analysis. Was it done manually or after computational analysis?

RESPONSE: *all granule identification was done manually. This is noted in the revised text (Lines 89-90).*

4) There are strong differences in signal intensity in some NanoSIMS imaging (Fig 3 for instance). Indeed, the signal observed in Fig. 3B or fig. 3D would have been included in the background signal in Fig. 3A or Fig. 3C. Is there any explanation for that? In addition, could it be possible to produce better quality superimposed images (center of Fig. 3, Fig. 4, Fig. 5) by deleting most of the background of NanoSIMS picture that darkens the picture? This would give a better view of the localization of the observed signal.

RESPONSE: *The reviewer makes an important point here for which there are several possible explanations. Firstly, it is possible that, between experiments, there are small differences in the amount of label supplied to the leaves. We do not believe this to be the case, since we always used pre-mixed, certified, ¹³CO₂-containing air. However, we do know from other experiments that there is biological variation in the assimilation rates from one plant to the next, but we do not have a measure of this in these experiments (indeed, ¹³CO₂ cannot be measured using conventional infra-red gas analysis). Second, the analysis is done on an individual chloroplast basis and the incident light and CO₂ reaching any given chloroplast may differ substantially, depending on the cell*

position (although we always tried to image the palisade cells), and the chloroplast position within the cell (this was not possible to control). Thus, one chloroplast may fix more ^{13}C than another, depending on its access to light and CO_2 . Thirdly, the signal intensity may also depend on the amount of glucan present in a given chloroplast. Even if the amount of photosynthetically fixed ^{13}C and the proportion allocated to starch biosynthesis were constant, any differences in the number of stromal pockets configured for starch production (in the case of granule initiation), or existing granule surfaces available for elaboration (in the case of granule growth) could lead to changes in signal intensity. We make reference to this in the revised manuscript (Lines 457-461). However, we prefer not to delete the background from the NanoSIMS images and hope that the presentation of the two images and the merge is sufficient for readers to see the enrichment pattern and its overlap with the cellular ultrastructure.

5) Fig. 6: a gbss- mutant has been used that shows a reduced incorporation of ^{13}C in the core of the starch granule (Fig. 6B) compared to the wild-type (Fig. 6A). Which kind of gbss- mutant is it? Is the GBSS protein still present within the granule but inactive, or a mutant without GBSS in starch? If the second option is correct, could it be that the reduced signal in the gbss- mutant is related to the lack of the protein within the granule? I do believe that ^{13}C incorporation from $^{13}\text{CO}_2$ is not limited to starch but could also be found in proteins or any other organic compounds that are synthesized during the light phase.

RESPONSE: The GBSS mutant used in this study is that described in Seung et al. (2015), as stated in the methods (Line 389) which lacks the GBSS protein (now stated (Line 273)). The reviewer is right that ^{13}C will eventually be found in proteins and other cellular structures synthesized during the light phase. However, we believe that the labelling experiment is too short for this to occur to any significant extent. The $^{13}\text{CO}_2$ supplied would need to be fixed via photosynthesis, make its way through metabolism to significantly enrich free amino acids in the cytosol. These would need to be incorporated into GBSS pre-proteins, re-imported into chloroplasts, processed, folded, and be targeted to - and enriched on - the starch granule. We consider it unlikely that this would produce a detectable NanoSIMS signal during the 60-minute experiment. Furthermore, even if detectable amounts of ^{13}C -labeled GBSS protein were produced and targeted to the granule during the experiment, it would presumably bind to the granule surface and contribute to the surface signal rather than appearing in the granule core, since only small molecules like ADPGlc can diffuse into the semi-crystalline amylopectin matrix. GBSS present in the cores of wild-type granules would have been made and incorporated into the granule prior to labelling. Hence, we do not believe that the label detected inside wild-type granules (but not in the gbss mutant) can be attributed to the GBSS protein itself.

6) Lines 327-329: the authors suggest that, since granule growth is anisotropic and preferentially observed at the equatorial regions of the granule, synthetic enzymes must be concentrated at the same region. This makes sense; however, it could also be that precursor (ADP-glucose) synthesis and/or concentration occurs preferably at the same region through a still not uncovered mechanism, thus privileging granule growth in specific direction.

RESPONSE: we appreciate this interesting and valid suggestion, which complementary and not mutually exclusive to our suggestions. We mention this option in the revised manuscript (Lines 341-342).

7) Lines 338: I understand the reference to fibrillins (FBN1a and FBN1b) that were shown to interact with SS4. However, none of the corresponding mutant display a phenotype that is related to that of the ss4- mutant. Thus, the actual implication of FBNs in the process of starch granule initiation is questioned.

RESPONSE: we agree with the reviewer on this point. While the previously published protein-protein interaction data seem valid, the lack of phenotypic impact upon mutating both FBN1a and FBN1b fibrillins calls the biological significance into question. We also mention in a prior publication that we could not reproduce this interaction. We have consequently de-emphasized this point by removing one reference to the fibrillins, without removing the concept altogether.

8) A comparative analysis was performed between Arabidopsis wild-type and mutant lines for SS4, which is known to affect the initiation process of starch synthesis *in vivo* by significantly reducing the number of granules per chloroplast and modifying their morphology. I think it would have been worth it to include a line with a phenotype opposite to that of the ss4- mutant. Indeed, the isa1- mutant would have been interesting. In that mutant, starch synthesis is altered but not stopped. Numerous small granules are synthesized in the isa1- mutant that could, somehow, correspond to the “granule initials” suggested by the authors.

RESPONSE: The reviewer suggests an interesting experiment. Indeed, we are pursuing further research in this direction, including mutants which show alterations in starch granule initiation frequency (e.g. ptst2 and the corresponding PTST2 overexpressing lines). The isa1 mutant would make an interesting addition. However, we respectfully suggest that this would better contribute to a second manuscript where we explore a number of recently described genetic factors in parallel.

9) Lines 310-319. I don't exactly understand the demonstration made here. What would be “opposing orientations of lamellae”? Why the lamellae would be unlikely radially oriented in granules initials? Do the authors have any evidence for that? By the way, if granule formation results from the merging of several “granule initials”, starch granules would then consist of several hila. I don't remember that Arabidopsis starch granules have been described containing several hila. Moreover, how does the growth ring would be organized in that case? Starch growth rings are organized around the hilum of the granule as already described elsewhere. This raises the question of how growth rings will organize if several “granule initials” merge to form one bigger granule. One last thing, which is about semantic: to my point of view, the 9-10 nm lamellae are not “radially arranged in starch”. This is the molecules inside the 9-10 nm lamellae that are arranged radially (the glucans are more or less oriented parallel to the radius of the starch granule). The lamellae are arranged tangentially to the surface of the granule (i.e. they are placed concentrically from the center (hilum) of the granule).

RESPONSE: we appreciate the reviewer pointing out that this speculative part of the manuscript is unclear and we have rewritten it to more clearly describe our ideas (Lines 322-337). We also appreciate the correction about it being the amylopectin molecules - not the lamellae – that are radially arranged, which was incorrectly phrased at one point.

Reviewer #2

RESPONSE: *We are glad that Reviewer 2 found the manuscript well structured and presented. We have added the requested details as described below:*

1) more data is needed for explanation of growing conditions, eg. how the authors have provided that test plants were not exposed to water/nutrient stress?

RESPONSE: *We have added more details about the growth of the plants in the materials and methods, including a comment on watering and nutrients (Lines 384 and 386).*

2) Also, more information is needed for plant material sampling (eg. leaf from which position) and handling prior analyses

RESPONSE: *We give clearer details about the sampling in the materials and methods (Lines 399-400)*

line 425, reference stile is incorrect.

RESPONSE: *We have italicized the reference number.*

Reviewer #3 (Remarks to the Author):

RESPONSE: *We are glad that Reviewer 3 found our work to be innovative, of high quality and convincing. We have added the requested details as described below:*

One aspect that could perhaps be made a bit clearer is the time frame of starch granule formation. In figure 3D for example, only one starch grain is labeled, while the others are not. In this context the chase of 4 hours could perhaps be set in proportion to the duration of granule formation.

RESPONSE: *we have added further information about the time frame of granule formation and improved the text in relation to the interpretation of Figure 3D. We thank the reviewer for pointing this out because it is easy to assume that one should always see label in every granule in such a section. In fact, there are two good reasons why one might not – either a granule was initiated after the period of ¹³C labelling or the labeled center of the granule is not in the section that was imaged. The granules in Fig 3D are roughly 1 micron in diameter, whereas the section imaged is only 60-70 nm in thickness. Thus, in a given section, there is no guarantee of capturing the centers of all the granules. This is now stated in the revised manuscript (Lines 190-192). We are working towards 3D NanoSIMS analysis so as to allow this, but cannot add these preliminary investigations to this manuscript.*

Figure 1F would perhaps benefit from an inset with higher magnification.

RESPONSE: we have provided Supplemental Fig. S1, which has further examples of this phenomenon shown at higher magnifications.

Reviewers' Comments:

Reviewer #1:

Remarks to the Author:

This is a revised version of the manuscript. It has been significantly improved (though the initial manuscript was actually of extremely high quality and fully innovative) and the authors have taken into account the suggestions/remarks made by the reviewers. I don't have any further comments, and I do believe that this work brings significant and important progress in our understanding of starch granule growth.

Reviewer #2:

Remarks to the Author:

The authors answered all my questions in details, as well as the inquiries of the other two reviewers. It is therefore my recommendation that the manuscript be accepted for publication.

Reviewer #3:

Remarks to the Author:

The revised version of the manuscript contains all additional explanations and details requested. I recommend publication of this interesting piece of work in its current form.